# Improvement of the Quality of Recycled Concrete Aggregate Subjected to Chemical Treatments: A Review

**DOI:** 10.3390/ma15082740

**Published:** 2022-04-08

**Authors:** Javier A. Forero, Jorge de Brito, Luís Evangelista, Cláudio Pereira

**Affiliations:** 1Postgraduate Program in Structural Engineering and Construction (PECC), Predio SG-12 Campus Darcy Ribeiro, University of Brasília, Brasilia-DF, CEP 70910-900, Brazil; jaaandresforero@gmail.com (J.A.F.); claudiopereira@unb.br (C.P.); 2CERIS, Instituto Superior Técnico, Universidade de Lisboa, Av. Rovisco Pais, 1049-001 Lisboa, Portugal; jb@civil.ist.utl.pt; 3CERIS, ISEL-IPL, Department of Civil Engineering, Instituto Superior de Engenharia de Lisboa, Lisbon Polytechnic Institute, Rua Conselheiro Emídio Navarro, 1959-007 Lisbon, Portugal

**Keywords:** mortar, pre-soaking treatment, recycled concrete aggregate, treatment, aggregate quality

## Abstract

The main factor that alters the quality of recycled concrete aggregate (RCA) is the paste adhered to the natural aggregate (NA). Since it causes weakening of the interfacial transition zone (ITZ) between the aggregate and the cementitious paste, it becomes a determining factor for the mechanical behavior of concrete. It turns out that it is critical to enhance this interface by improving the surface of the aggregate or by removing the paste adhered to the NA. Considering the variety of methods for removing paste adhered to RCA—namely using acids such as hydrochloric acid (HCl), sulfuric acid (H_2_SO_4_), and phosphoric acid (H_3_PO_4_), among others—this paper presents a review of treatments for the removal of adhered paste using acidic solutions on the RCA, and their influence on the mechanical properties and durability of concrete produced with RCA. Pearson’s correlation was used in the statistical analysis to determine the linear relationship of the main factors—for instance, immersion time, acidic solution, and aggregate size—involved in the removal of the paste in the RCA.

## 1. Introduction

Within construction, the use of recycled materials is being promoted in order to obtain new concrete. Among these materials, recycled concrete aggregate (RCA) is used as a substitute for natural aggregate (NA). In spite of the possible environmental benefits, these new materials face limitations to achieving conventional performance. Compared to NA, RCA shows high porosity, high water absorption, low density, and weak adhesion between the interfacial transition zone (ITZ) and the contained mortar, resulting in a low mechanical behaviour of the concrete [1,2,3,4]. Therefore, the modification of the microstructure of the ITZ has been a major concern for improving the properties of recycled aggregate concrete (RAC).

Two main techniques are identified to improve RCA. The first one is the elimination of the paste adhered to the RCA by means of chemical treatments [5,6,7,8,9,10,11,12,13,14,15], thermal treatments (traditional heating, microwave heating) [7,16], mechanical treatments [17,18], the combination of heating and mechanical treatments [19], and the combination of chemical and mechanical treatments [5,20]. The second technique is the use of polymers to enhance the interaction between RCA and cement paste [21]. The addition of silicones, calcium carbonate bio-deposition, and pozzolanic materials—such as fly ash, silica fume, carbonation and silicate sodium solution—are part of this technique.

The aim of this article is to describe the current scientific state-of-the-art of the characterization and use of RCA treated with different acidic solutions, and to consider the possible advantages that this type of treatment brings to RCA, mainly on its mechanical behavior and durability properties. Environmental benefits based on the reuse of acid mine drainage (AMD) may also be considered [22]. The first part of this article explains the chemical process of removing the mortar with most-often-used chemicals, i.e., hydrochloric acid (HCl), sulfuric acid (H_2_SO_4_), and phosphoric acid (H_3_PO_4_). Then, improvements to concrete in the fresh and hardened state are presented, ending with a Pearson’s correlation statistical analysis of several variables (acid molar concentration, immersion time, aggregate size, and acid solution) to determine its degree of influence on the removal and water absorption of RCA.

## 2. Chemistry of Acid Treatment in RCA

The chemical treatment technique of pre-soaking RCA in acidic solutions has been an effective method for the elimination of adhered mortar in RCA and the improvement of its properties. It consists of an acid attack associated with the reaction of an acid (HX) and the calcium hydroxide (CH) portion of the cement paste; this produces a highly soluble calcium salt (CX, where X is the negative ion of the acid) by-product which is easily removed from the cement paste, thus weakening the paste’s structure as a whole (Equation (1)) [23,24].
(1)HX+CH→CX+H

Once the acid interacts with concrete, three major reactions are triggered. Firstly, the hydration products react with acid, forming dissolved ions, thus losing the mortar adhered to the NA; this reaction is called acidolysis. Secondly, depending on the type of acid, insoluble salts form and precipitate, causing expansion and cracking. Other reactions create complexes with calcium, aluminum, iron, and silicate ions, which produce much higher concentrations of these ions in the solution that would otherwise lead to dissolution. This can potentially occur under pH conditions in which cement would normally be relatively stable (complexolysis). Depending on the acid used, more than one of these detrimental mechanisms can be effective [25,26].

The first mechanism increases the content of sulfate ions (SO_4_^2−^) in the adhered mortar of the RCA through the addition of HCl, H_2_SO_4_, and H_3_PO_4_. The chemical reactions responsible for cement corrosion depend on the type of acidic solution. Cement is an alkaline material with C_3_S, C_2_S, C_3_A and C_4_AF as its main components, whose reaction with water form hydration products—calcium silicate hydrate (C-S-H), calcium hydroxide or portlandite (CH), ettringite (C_6_AS_3_H_32_), calcium monosulphoaluminate (C_4_AS_3_H_12_), and hydrogarnet (C_3_AH_6_), among others; thus, they constitute a cementitious structure that can easily be attacked by strong acids [27,28].

A sulfate attack generates a series of reactions in the cementitious matrix of the concrete, due to the penetration of SO_4_^2−^ in the cement structure from the exposure of the RCA to acids and the environment. When external sources of sulfate ions penetrate the cementitious matrix, initiating chemical reactions with the hydrated products such as portlandite (Equation (2)), they form gypsum and calcium aluminate phases, which form ettringite; this is followed by reactions of monosulfate (Equations (3) and (4)), tricalcium aluminate (Equations (5) and (6)), tetracalcium aluminate hydrate (Equation (7)) and hydrogarnet (Equation (8)) [29,30]. The calcium ions are initially supplied by portlandite, and when the ions are no longer available, the calcium silicate hydrate dissociates in the silica gel, providing the ions for the formation of ettringite [30,31].
(2)$${\mathrm{Na}}_{2}{\mathrm{SO}}_{4}+\mathrm{CH}+2H\to {C\bar{S}H}_{2}+2\mathrm{NaOH}$$
(3)$$C_{4}{A\bar{S}H}_{12}+2{C\bar{S}H}_{2}+16H\to C_{6}A\bar{S_{3}}H_{32}$$
(4)$$3C_{4}{A\bar{S}H}_{12}+3{\mathrm{Na}}_{2}{\mathrm{SO}}_{4}\to 6\mathrm{NaOH}+2\mathrm{Al}{\left(\mathrm{OH}\right)}_{3}+21H+2C_{6}{A\bar{S}H}_{32}$$
(5)$$C_{3}A+3{C\bar{S}H}_{2}+26H\to C_{6}A\bar{S_{3}}H_{32}$$
(6)$$C_{3}A+3{\mathrm{Na}}_{2}{\mathrm{SO}}_{4}+3\mathrm{CH}+32H\to 6\mathrm{NaOH}+C_{6}A\bar{S_{3}}H_{32}$$
(7)$$C_{4}{\mathrm{AH}}_{13}+2{C\bar{S}H}_{2}+14H\to C_{6}A\bar{S_{3}}H_{32}+\mathrm{CH}$$
(8)$$C_{3}{\mathrm{AH}}_{6}+3{C\bar{S}H}_{2}+2\mathrm{OH}\to C_{6}A\bar{S_{3}}H_{32}$$

The acidolysis mechanism takes place when the cement hydration products react with acids in the form of dissolved ions [25]. In particular, when the acidic solution penetrates the pores of concrete, it dissolves the constituents; calcium cation are the first to be dissolved (Figure 1), given that portlandite becomes soluble at high pH values. Exposure to the acid solution also causes loss of calcium—descaling—from the C-S-H gel, leaving behind a relatively weak silica gel. These poorly soluble salts can act as a partial inhibitor of the general process, blocking the tiny passage of cement paste through which water flows [23,25].

Consumption of cement hydration products, particularly Ca(OH)_2_ due to the action of HCl and nitric acid (HNO_3_), results in the formation of CaCl_2_ and Ca(NO_3_)_2_ salts, respectively. These soluble salts can be easily transported to the external parts of mortar using water. In this situation, the continuous reactions increase the porosity of the cement paste, and the increase in pore volume accelerates the reaction rate. In the case of an attack of H_2_SO_4_, the assessment of deleterious reactions can be divided into two parts. In the first stage, the deterioration of Ca(OH)_2_ results in expansive plaster formation. Then, the plaster reacts with C-S-H in an aqueous environment and forms a more expansive product called ettringite [20,32].

Another chemical reaction present in the removal of adhered mortar in RCA is the one between the mineralogical nature of the RCA and acids. According to Dyer [33] when limestone aggregate is in contact with an acid, the following reaction occurs (Equation (9)):(9)$${\mathrm{CaCO}}_{3}+2\mathrm{HX}\to {\mathrm{CaX}}_{2}+2{\mathrm{CO}}_{2}$$

Limestone aggregates neutralize acids as they dissolve, making it difficult to remove mortar from the RCA, whereas siliceous aggregates are inert to acid attack [34,35].

## 3. RCA Properties

The mechanical and durability properties of concretes produced with RCA are directly influenced by the characteristics of the source material, e.g., w/c ratio, type of cement, and strength, in addition to other factors such as production and storage [36,37,38,39]. This high variability in the RCA can affect the removal of adhered mortar using acids. RCA made with cement with a high C_3_S content show higher resistance to compression at 28 days, demands a higher w/c ratio, and hydrates faster. The abovementioned results in a higher content of Ca(OH)_2_ and a larger porous structure that is more vulnerable to an acid attack, increasing the removal of adhered mortar [40]. The effectiveness of removing mortar in RCA from concretes with a partial replacement of cement with pozzolanic admixtures or chemical admixtures varies due to differences in their chemical composition, i.e., the type of cement or chemical admixtures used. Pavlík [41] investigated the influence of the w/c ratio on the mortar corrosion process by acetic acids (CH_3_COOH) and HNO_3_, showing that the corrosion rate decreases with an increase in the cement content per unit volume of the hardened paste. The author stated that there are two main causes related to this fact: the increased neutralization capacity of the matrix and the increase in diffusion resistance of the corroded layer. The following are the main properties of RCA after being treated with acid solutions.

### 3.1. Water Absorption

Owing to its porous structure, the paste adhered to RCA is more susceptible to water absorption than NA and needs extra water during mixing. A failure to compensate this water will negatively affect the concrete’s performance [42,43]. There is a high dispersion of water absorption values in RCA, ranging from 1% to 20% [37,44,45,46,47], depending on particle size and density. There are several techniques to improve or modify these characteristics. One of the main techniques for the removal of mortar in RCA is pre-soaking in acidic solutions.

The technique initially proposed by Tam et al. [15] consists of the immersion of RCA in acidic solutions at different concentrations and immersion times, in order to eliminate the mortar adhered to NA. Afterwards, the RCA is washed with water and oven-dried for 24 h at 100 °C, which allows it to be used. This technique is easy to implement and does not require specific equipment. Tam et al. [15] superficially treated RCA in acidic solutions at low concentrations of 0.1 mol with HCl, H_2_SO_4_, and H_3_PO_4_; in addition, RCAs with particle sizes of 20 and 11 mm were soaked for 24 h at 20 °C and the decrease in water absorption was obtained: 12.16% for HCl-treated RCA, 11.0% for H_2_SO_4_-treated RCA, and 8.36% for H_3_PO_4_-treated RCA for size 11 mm; the results were similar for a particle size of 20 mm, with a reduction in water absorption of 12.12% for HCl-treated RCA, 10.3% for H_2_SO_4_-treated RCA, and 7.27% for H_3_PO_4_-treated RCA. The different acids increase the content of SO_4_^2−^ in the adhered paste of RCA; when reacting with the main mortar compounds of RCA, as described in Section 2, this allows the removal of the adhered mortar, reducing the water absorption capacity of RCA.

Akbarnezhad et al. [7] showed that acid treatment at low concentrations reduces RCA’s water absorption. Nevertheless, none of these values approach those of NA. Akbarnezhad et al. [7], used H_2_SO_4_ to remove the bonded mortar in NA at concentrations of 0.1 M, 0.5 M, and 1 M, with a pre-soaking time of 1 and 5 days. The water absorption reductions of RCA were 2.3% for a 0.1 M molar concentration soaked for 24 h and 120 h, and 16.6% and 61.9% for 1 M molarity submerged for 24 and 120 h, respectively. Additionally, greater effectiveness occurred when using higher acid concentrations.

Ismail and Ramli [9] studied the immersion of RCA in low HCl molarities of 0.1 M, 0.5 M, and 0.8 M, with pre-soaking times of 1, 3, and 7 days to improve RCA’s characteristics. The study showed that water absorption was reduced in the range of 1% to 28%. Greater reduction was observed at 0.5 M and 0.8 M concentrations and in smaller RCA particle sizes. In addition, it was observed that the pre-soaking time of RCA in the acidic solution was not a determining factor in reducing water absorption. Ismail and Ramli [10], who analyzed the effect of using HCl in a 0.5 M concentration solution for a period of 24 h, reported a reduction in the water absorption of RCA of between 19.3% and 16.6%, relative to untreated RCA.

Purushothaman et al. [13] observed that RCA treated with HCl and H_2_SO_4_ in five particle sizes (20.0, 16.0, 12.5, 10.0 and 4.75 mm), at a concentration of 0.1 M and pre-soaked for 24 h, had a reduction in water absorption of 41% and 58%, respectively, for HCl and H_2_SO_4_, compared to untreated RCA.

Acid treatments of RCA have also been explored by Pandurangan [12], who studied the effects of the bond between acid-treated RCA concrete and reinforcement. Nitric acid (HNO_3_) was used to pre-soak RCA at a 1 M molarity for 24 h, a procedure described by Movassaghi [48]. The results obtained were a 36.7% reduction compared to untreated RCA.

Saravankumar et al. [14] also improved the water absorption of RCA using HCl, H_2_SO_4_, and HNO_3_, at a concentration of 0.1 M and after pre-soaking for 24 h; they showed that there was an improvement in water absorption of 10%, 11% and 13% for HCl, HNO_3_ and H_2_SO_4_, respectively.

Al-Bayati et al. [8] also analyzed the effect of acidic solutions of HCl and acetic acid (C_2_H_4_O_2_) on RCA, treated at a concentration of 0.1 M and with a 24 h soaking time. There was a decrease in water absorption of 4.22% for HCl, and of 4.33% for C_2_H_4_O_2_.

Kim et al. [11] also studied the effect of acid treatment on RCA, by treating RCA with HCl and sodium sulfate (Na_2_SO_4_) in an acidic solution at a 1:4.5 aggregate: acid ratio (1.2 M), submerged for 48 h, and replacing the solution with a new one after 12 hours of pre-soaking. The results showed a reduction of 38.6% for HCl and of 34.9% for Na_2_SO_4_.

Tang et al. [49] treated RCA using a 0.5 M solution of H_2_SO_4_ by pre-soaking it for 24 h and shaking the RCA occasionally inside the acidic solution. Then, to ensure that the treated RCA has no residue from the acidic solution, the RCA was washed and submerged in water for 24 h. The results after treatment showed a 10% reduction in water absorption compared to untreated RCA.

Wang et al. [20] pre-soaked RCA in an acetic acid (CH_3_COOH) solution at an ambient temperature at three different acid concentrations (1%, 3%, 5%), and three different immersion durations (1, 3, and 5 days). The water absorption of all RCA samples was reduced by 9–19%. The lowest water absorption performance was achieved for RCA treated with 1% acetic acid. The authors mentioned that greater incorporation of acetic acid increases water absorption of the treated RCA. The authors also explained that this occurs mainly because more pores were produced in the treated RCA samples due to the dissolution of more hydration products, and possibly some NA in the acetic acid.

### 3.2. Determination of Mortar Loss

The percentage of mass adhered to RCA is determined by the difference between the initial weight and the final weight, as shown in Equation (10). Ismail and Ramli [9] studied the effect of HCl acid on RCA at different concentrations (0.1 M, 0.5 M, and 0.8 M) and immersion times (1, 3, and 7 days). The results showed that at high concentrations, the loss of mass is higher, with a linear correlation between these two factors; moreover, the soaking time did not have a great influence on removed mortar mass. Ismail and Ramli [9,10] also showed that removal is greater at smaller RCA sizes. This can be explained by Juan and Gutierrez [50] who removed the mortar adhered to RCA using HCl acid, showing that smaller aggregates have a range of adhered mortar of around 33% to 55%, while larger fractions range from 23% to 44%. Other authors also report that the content of adhered mortar is higher for smaller RCA sizes [5,51]. The removal of the paste adhered to RCA is more efficient in aggregates with a higher mortar content, considering that the removal of the mortar depends on the type of cementitious material used in the RCA [52]. Pavlik [53] showed that the rate of corrosion decreases with an increase in cement content per unit volume of hydrated cement paste (increases with w/c).
(10)$$\%\mathrm{Adhered}\;\mathrm{mortar}\;\mathrm{loss}=\frac{\mathrm{Mass}\;\mathrm{of}\;\mathrm{RCA}-\mathrm{Mass}\;\mathrm{of}\;\mathrm{RCA}\;\mathrm{after}\;\mathrm{treatment}}{\mathrm{Mass}\;\mathrm{of}\;\mathrm{RCA}}$$

A more aggressive and destructive case of acid attack occurs when concrete is exposed to sulfuric acid. The calcium salt produced by the reaction of the H_2_SO_4_ and calcium hydroxide is calcium sulfate which, in turn, causes increased degradation due to a sulfate attack.

Akbarnezhad et al. [6] used the H_2_SO_4_ treatment technique to determine the amount of paste adhered to RCA, and proposed four techniques for total removal of the mortar: the first one was to submerge the RCA in H_2_SO_4_ at concentrations of 1 M to 6 M; the second technique consisted of submerging the RCA in H_2_SO_4_ at concentrations of 1 M to 6 M, renewing it after every 8 h of immersion, after which the RCA was washed before submerging it again in the solution; the third technique was to submerge the RCA at the same concentrations but with continuous agitation of the particles; finally, the fourth technique considered all the procedures previously described in the second and third cases. The results obtained showed that the removal of mortar varied between 12% and 100%, showing that the main factors for this removal were acid concentration and the removal technique (best results: techniques II and IV). In addition, it was observed that the acid concentration lost H^+^ ions over time, reducing the degree of attack of the mortar. Washing the RCA before each replacement removed a layer of silica and aluminosilicate gels released by C-S-H from the RCA surface [52], increasing the removal efficiency.

Akbarnezhad et al. [7], used H_2_SO_4_ to remove the mortar at concentrations of 0.1, 0.5 and 1 M, reporting mass-losses of 2%, 14%, and 34% after one day of exposure, and mass-losses of 2%, 13% and 34% after 5 days of exposure, respectively. Authors such as Al-Bayati [8] and Saravankumar et al. [14] reported mass-losses of 3.92% and 5% for RCA for one day of soaking at concentrations of 0.1 M and 2.7 M, respectively.

Pre-soaking in acid to determine the mortar loss was also investigated by Kim [11], who studied the effect of using HCl and Na_2_SO_4_. The author obtained mass-losses between 2.66% and 9.09% for RCA sizes from 2.36 mm to 19 mm for Na_2_SO_4_. The values for HCl were 5.34% to 19.91% for the same aggregate sizes. The biggest removal was for HCl, as it is a more aggressive acid.

### 3.3. Bulk Density

Akbarnezhad et al. [6] produced two types of concrete with 30 MPa and 60 MPa of compressive strength for the production of RCA, using only particles between 8 and 12 mm. There was a linear correlation between the content of mortar in RCA and the decrease in bulk density for the two types of aggregate. The results showed that an increase in mortar content from 0% to 52% (by mass) for RCA produced from 30 MPa concrete resulted in an almost linear decrease in the apparent density of RCA from 2590 to 2295 kg/m^3^. Similarly, an increase in mortar content from 0% to 58% (by mass) for RCA produced from 60 MPa concrete led to a linear decrease in the bulk density of RCA from 2590 to 2340 kg/m^3^. The results showed that, as the density of mortar in 60 MPa concrete is higher than that of 30 MPa concrete, a similar mortar content in RCA produced from 60 MPa concrete has a higher bulk density than RCA produced from 30 MPa concrete.

Ismail and Ramli [9] showed that untreated RCA has lower physical properties than NA, and that the particle densities (dry oven condition) of 20 mm and 10 mm RCA were 2330 kg/m^3^ and 2230 kg/m^3^, respectively. These figures were lower than those of 20 mm and 10 mm NA (2600 kg/m^3^ and 2580 kg/m^3^, respectively). However, the physical properties of RCA improved after immersion in acid, with a higher increase for 10 mm aggregates (4.7%) than for 20 mm aggregates (2.2%) owing to the content of adhered mortar tending to be greater in smaller aggregates than in coarser aggregates [50]. Due to the relationship between aggregates’ density and absorption [37,44,54], the increase in RCA density results in the significant decrease in RCA water absorption. The density of RCA increases at varying concentrations of acid treatment.

Al-Bayati [8] also saw improvements of 7.37% and 5.40% in bulk density for C_2_H_4_O_2_ and HCl, compared to untreated RCA. Saravanakumar et al. [14] observed that, after RCA treatment using three acids (H_2_SO_4_, HNO_3_ and HCl), the bulk density had variations of less than 10%, 13% and 13% for each acid compared to untreated RCA, which showed a variation of 15% in relation to NA.

### 3.4. Microscopic Analysis of the RCA

According to Ismail and Ramli [9,10], acid treatment at low molarities (0.1 M and 0.5 M) with HCl in RCA reduces the number of loosely adhered particles, leaving a cleaner, smoother surface with less-sharp angles [44]. The authors also observed that, with the increase in molar concentration, the brittle particles of the mortar are released due to the increased attack on the mortar caused by the greater addition of H^+^. Figure 2 shows a change in the morphology of RCA before (Figure 2a) and after treatment with HCl at molar concentrations of 0.1 M (Figure 2b), 0.5 M (Figure 2c), and 0.8 M (Figure 2d). A scanning electron microscope (SEM) clearly shows cleaner and more uniform surfaces. Saravanakuar and Manoj [14] also analyzed the microstructural morphology of the effects of treatments with HCl, HNO_3_ and sulfuric acid on the surface of the RCA, showing the same degradation of the paste observed by Ismail and Ramli [9,10]. Al-Bayati et al. [8] showed that there are differences on the surface when RCA is treated with strong acids than with weak ones, reaching the same conclusion as the previous authors. A summary of the techniques discussed in this section is presented in Table 1.

## 4. Properties of Concrete with Treated RCA

### 4.1. Fresh-State Properties and Density

The properties of the fresh state of RCA-containing concrete are directly affected by its presence, as it has sharper geometries that reduce the slip between particles and increase water absorption, decreasing the workability of the concrete. Controlling workability is important to achieve adequate compaction [27]. The main results of the influence of chemical treatments on the production of concrete with RCA are shown below.

Ismail and Ramli [9,10] produced concrete with various ratios of RCA incorporation (15%, 30%, 45%, 60%) treated with HCl at molar concentrations of 0.1 M, 0.3 M, and 0.8 M with pre-soaking times of 1, 3, and 7 days, and noted that workability did not show significant differences: 17% higher in concrete produced with treated RCA than in concrete with untreated RCA. This result may be attributed to the high absorptivity of coarse RCA caused by the porous mortar attached to it, which absorbs more water during concrete mixing, thus lowering the workability of concrete. Ismail and Ramli [9] also observed that workability decreases linearly with the increase in the incorporation ratio of acid-treated RCA. The author also reported that the density of concrete made with treated and untreated RCA has no significant difference. In fact, the density of concrete with treated RCA is slightly higher due to the reduction in the adhered mortar to the original NA. This behavior was also observed by Al-Bayati et al. [8], who treated RCA with HCl and C_2_H_4_O_2_ at 0.1 M and a 24-h pre-soaking time. The authors also noted that there is a tendency for density to decrease when the incorporation ratio of treated RCA increases. This trend is also the same as other reported by other authors when using untreated RCA in the manufacture of concrete [55,56].

Purushothaman et al. [13] saturated RCA after treatment with HCl and H_2_SO_4_ before the production of concrete, and showed that concrete with treated RCA had an improvement in workability of around 23.5% for both acids relative to concrete with untreated RCA. Similar results were recorded by Pandurangan et al. [12] and Butler et al. [57] in concrete made with RCA treated with HNO_3_.

Similar results were obtained by Kim et al. [11] as they observed an improvement in the workability of acid-treated RCA concrete. The authors noted that workability improves when RCA is treated with strong acids (HCl), as they have a greater degree of removal than weak acids, leaving a less angular surface that promotes fluidity due to a ball bearing effect [3,34].

In summary, it was observed that all authors reported insignificant increases in the workability of concrete mixes containing RCA treated with acid solutions of low molarities. The differences are due to the fact that mortar adhered to NA still remains.

### 4.2. Compressive Strength

As previously mentioned, concrete with RCA presents a weaker interfacial behaviour. Tam et al. [15] tested several mixes with coarse RCA treated in acid solutions of HCl, H_2_SO_4_, and H_3_PO_4_, and substitution ratios of 5%, 10%, 15%, 20%, 25%, and 30%. The authors showed that the compressive strength after treatment had significant improvement, i.e., increases of 10.1%, 11%, 0,07% 6,67%, 12.8%, and 14.0% for HCl, H_2_SO_4_, and H_3_PO_4_, respectively, and incorporation ratios of 20% and 25% of RCA.

Ismael et al. [9] studied several mixes incorporating coarse RCA at 15%, 30%, 45%, and 60% replacement ratios, treated with HCl at molarities of 0.1 M, 0.5 M and 0.8 M, a water/cement ratio (w/c) of 0.4, a cement content of 537 kg/cm^3^, and a 28-day compressive strength of 50 MPa. The results presented by the authors showed that the maximum replacement level to maintain strength at 28 days is up to 45% for RCA treated with HCl at molarities of 0.1 M and 0.5 M and not 0.8 M; this is because high concentrations are detrimental to the surface of RCA, leaving it more brittle and fragile and interfering with the good connection between cement and the particles. It was also observed that ratios of 15% incorporation of RCA in the mixes result in improvements in compressive strength, relative to mixes that incorporate untreated RCA. On the other hand, the authors showed that the effect of different RCA soaking ages on the compressive strength is insignificant.

A study by Ismail and Ramli [10] determined the influence of RCA treated with HCl at 0.5 M, submerged for 1 day, on its mechanical and drying properties. The characteristics of the mixes were: a w/c ratio of 0.40, a cement content of 510 kg/m^3^, a 28-day compressive strength of 50 MPa, and a 60% replacement ratio of NA with treated RCA. The authors determined the compressive strength of concrete at 7, 28, 90, and 180 days. Concrete prepared with treated RCA had better performance than that prepared with untreated RCA. At 7 days, treated RCA concrete had a compressive strength 3% higher than that of control concrete. At 28, 90 and 180 days, the compressive strength of concrete with treated RCA was 96%, 99%, and 98% that of the control concrete, respectively, i.e., almost the same.

Ismail and Ramli [10] determined the ultrasonic pulse velocity (UPV) in mixes containing treated RCA and untreated RCA. They found that the UPV values are slightly higher in mixes with treated RCA. This is due to the improvement of RCA properties, thus improving the ITZ microstructure and increasing the bond strength between the new cement paste and RCA.

Purushothaman [13] studied various concrete mixes, including 100% coarse RCA treated with two types of acid—HCl and H_2_SO_4_—at a concentration of 0.1 M, soaked for 24 h, with a w/c ratio of 0.45, containing 380 kg/m^3^ of cement, and under a 28-day compressive strength of 30 MPa. It was observed that the mixes with untreated RCA reached 80% of the reference mix’s 28-day compressive strength, while the mixes with treated RCA reached 90% and 95% of the same value for HCl and H_2_SO_4_, respectively.

Saravanakumar et al. [14] studied the effect on concrete of RCA treated with H_2_SO_4_, HNO_3_, and HCl designed according to the ACI method [58], and mixing ratios of 1:1.4:2.3 (cement:sand:gravel) were used with a w/c ratio of 0.45. Ordinary Portland cement ASTM type 1, with a specific surface area of 3960 cm^2^/g and specific gravity of 3.15, was used. The authors replaced NA with RCA at 100%. The 28-day compressive strength of concrete made with recycled aggregates was 25% lower than that of concrete made with NA aggregates. In the treated aggregates, the loose mortar was removed as much as possible, and the characteristics of the aggregate’s surface were improved. The contact in the ITZ between the treated RA and the new cement paste improved and; thus, the 28-day compressive strength of concrete improved the treated RCA by 8 to 18% compared to concrete made with untreated RAC. The compressive strength development in concretes of RCA treated at later ages, between 28 and 91 days, was considered good. The relative strength development of concrete with recycled aggregate treated with HCl, HNO_3_ and H_2_SO_4_ was 18%, 18.5%, and 20%, respectively, at the age of 91 days. Among all the treated aggregate mixes, the development of strength was lower for the one treated with HCl.

Pandurangan et al. [12] performed mixes of concrete with RCA to assess the pull-out force between steel and concrete. The mixing ratio by weight was designed according to the ACI method [58] as 1:2.18:2.82 (cement:sand:gravel), with a cement content of 380 kg/m^3^, and a w/c ratio of 0.45 for concrete class M35. The replacement ratio was 91.5% according to the design method established by Fathifazl et al. [59]. The authors showed that the compressive strength improves by treating the recycled aggregates and represents more than 95% that of concrete with NA only.

Kim et al. [11] obtained results that are in accordance with the existing literature on mechanical properties of concrete with RCA. The compressive strength decreased when natural aggregates were replaced with recycled aggregates. Concrete that incorporated treated RCA exhibited better strength attributes than that with untreated RCA. Compressive strength increased by about 4.5%. This showed that pre-treatment of RCA with HCl is beneficial to the mechanical performance of concrete with RCA.

Wang et al. [20] showed that all concretes containing RCA treated with 1% and 3% acetic acid have higher compressive strength compared to the reference concrete. The strength increased more than 25% at 28 days in concretes with treated RCA. Wang et al. [20] showed that during the process, in higher acid solutions, the residual effect of the acid negatively influences the compressive strength with a greater influence at the ages of 3 and 7 days. Another possible reason for this reduction was indicated by the potential damage induced to the RCA.

The removal of mortar adhered to RCA is not limited only to the coarse aggregates. Kim et al. [60] studied the effect on the compressive strength of mortars manufactured with fine aggregates of RCA treated with H_2_SO_4_ and HCl. Unlike the trends of previous authors, they obtained lower results for compressive strength in mortars treated with acidic solutions. This behavior was explained by the production of gypsum in the removal, which, if not correctly eliminated, would come into contact with the calcium alumina of the cement, generating a greater number of voids.

Figure 3 shows the compressive strength results of several authors [9,10,11,13,14] on concretes with aggregates treated with HCl solutions at different ages of curing. A correlation of R^2^ = 0.70 between the different results can be observed. This variability is due to the fact that the removal is directly linked to the type of cement and the aggregate size. The results also show that there is a positive evolution in the compressive strength when the acid concentration is increased. This evolution in the compressive strength can be affected by the ionization of the acid increasing the content of ions in the RCA [60,61,62,63]. It also presents a confidence interval of 95% (*p*-Value = 0.0001) for the response in compression with this type of treatment.

### 4.3. Tensile Strength

Tam et al. [15] reported increases in tensile strength in concrete containing acid-treated RCA. They stated that the best performance improvements in 28-day flexural strength were 4.44% for 25% incorporation of RCA treated with HCl, 5.10% for a 10% RCA treated with H_2_SO_4_ and 18.58% for 10% RCA treated with H_3_PO_4_.

Ismail and Ramli [10] also reported improvements in the flexural strength of concrete containing treated RCA versus untreated RCA. For treated RCA concrete, the reduction was 4%, 9%, 3%, and 5% relative to the reference concrete at 7, 28, 90, and 180 days, respectively; this indicates that the treatment of the RCA with acid improved the flexural strength, because the reduction without treatment was 3%, 12%, 10%, and 13% at the same ages. Another important result is the good linear correlation that exists between flexural strength and compressive strength, which had a Pearson’s coefficient of R^2^ = 0.88.

### 4.4. Modulus of Elasticity

The modulus of elasticity (E) of concrete is directly related to the characteristics of the materials that compose it. RCA can be treated as a composite material of NA and mortar; therefore, the E of RCA is related to the E of the mortar (low) and to that of NA (high), in addition to other factors such as the resistance of the ITZ, stiffness, porosity and volumetric fraction of the paste in the RCA [64,65,66], resulting in concrete with lower E values than those of mixes produced with NA only [55].

Tam et al. [15] showed that E had significant improvements when using treated RCA. These improvements reached up to 20.4% (30% RCA treated with HCl), 15.37% (10% RCA treated with H_2_SO_4_) and 10.82% (30% RCA treated with H_3_PO_4_) in relation to the reference concrete at 7 days. The improvement was related to the amount of adhered mortar removed using the different acidic solutions, which led to a less porous aggregate with less-fragile particles.

Ismail and Ramli [10] also found improvements in E values for mixes produced with treated RCA. The mixes with treated RCA had reductions of 4%, while the ones with untreated RCA showed reductions of 12%, in both cases, when compared to the corresponding reference mixes. These improvements were due to the treatment effects mentioned in the previous paragraph.

Purushothaman et al. [13] observed that, in mixes with untreated RCA, the reduction in E was almost 35% relative to the reference concrete, while in mixes treated with HCl and H_2_SO_4_, the reductions in E relative to the mixes with natural aggregates were less than 8% and 22%, respectively. This better behaviour was due to the fact that the RCA had an increase in density, a removal of adhered mortar, and an improvement in the ITZ between the RCA and the cementitious paste.

### 4.5. Shrinkage

Studies carried out with RCA on its shrinkage reported a significant increase in the shrinkage of mixes with RCA, compared to conventional concrete.

From the results of shrinkage, Ismail and Ramli [10] observed that the shrinkage of mixes with treated and untreated RCA, which was measured for up to 180 days in total, fell below a micro strain of 500. From the results at the early ages (up to 28 days), the drying shrinkage behavior of all concrete mixes with untreated RCA was observed to be extremely steep (Figure 4). After 28 days, however, the drying shrinkage measured after 180 days for concrete with untreated RCA was 26% greater than that of the control concrete. The high shrinkage of concrete with untreated RCA was related to the low quality and low stiffness of the untreated, coarse RCA. The volume and stiffness of the aggregates are considered important factors that prevent the shrinkage of concrete [67].

### 4.6. Chloride Ion Penetrability and Carbonation Resistance

The results presented by Kim et al. [11] showed the permeability to chloride ions of untreated RCA mixes to be 27% higher than that of the mixes with NA only. Concerning the permeability to chloride ions of mixes containing RCA treated with HCl and Na_2_SO_4_, the penetration was 11% and 14% higher, and those with NA observed an improvement. The carbonation depth of the mixes followed similar trends to those in the chloride penetration tests. The carbonation depth of cast concrete increased by 9% compared to the concrete made with NA. This clearly shows the reduced durability characteristics of this mix when subjected to severe exposure conditions. According Kim et al. [11] the greater depth of carbonation is attributed to the limited formation of C-S-H gel during hydration, resulting in greater porosity and a less dense matrix.

### 4.7. Interfacial Zone between Cement Paste and RCA

As previously described, the quality of the ITZ depends on the characteristics of the RCA surface. This topic presents ITZ improvement differences between the treated and untreated RCA. Tam et al. [15] performed SEM on concrete mixes with treated and untreated RCA. In Figure 5a, an ITZ with a less dense bond between the paste and RCA is clearly observed. This behavior of the ITZ was also observed by Poon et al. [1], Sidorova et al. [2] and Xiao et al. [68]. When the RCAs were treated with HCl, H_2_SO_4_ and H_3_PO_4_, the ITZ was reduced (as seen in Figure 5b–d), resulting in a stronger bond between the aggregate and the cementitious paste; this was reflected in the improved mechanical properties of the concrete, as explained above.

Similar research has been conducted reporting the effectiveness of incorporating RCA treated with different acid solutions in concrete and mortar mixes. A summary of the different mechanical parameters that have been determined so far for concrete made with RCA treatment, and the type of acid solution, is provided in Table 2.

The results indicated that the properties of the treated RCAs depend mainly on the molar concentration of the acid solution used for treatment, the amount of cement paste adhered to NA, and the aggregates and water/cement ratio used.

## 5. Statistical Analysis

From the analysis of the collected data, it was observed that there are parameters that seem to be more influential in removing the mortar adhered to RCA. The main parameters observed from the literature review were molar concentration, pre-soaking time, type of acidic solution, and aggregate size. For this statistical analysis, two types of acid were chosen (HCl and H_2_SO_4_), because they are the ones most used by researchers. The analysis established the correlations that exist between the input parameters and the improvement after processing.

### Statistical Analysis of RCA Properties

Using Pearson’s correlation to determine the intensity and direction of the linear relationship between variables [69,70,71], an analysis was made regarding molar concentration, soaking time, RCA size and water absorption. From the analysis, it can be seen that there is a small number of relevant correlations. As shown in Table 3, only some correlations are significant, and they have a reduced degree of significance (*p*-value equal to or less than 0.05) [69,72]. The results for the HCl treatment of RCA show that Pearson’s correlation between molar concentration and mortar loss is r = 0.40, indicating a weak correlation between these two variables. This can be explained by the high variability in the mortar content of the RCA, which means that the same aggregate sizes can have different percentages of adhered mortar [50,73,74,75,76,77], as seen in Figure 6a. Another important parameter is the degree of significance between the two variables, *p*-Value = 0.05, with a 95% chance of loss of mass with an increase in acidic molarity. The results show that there is a linear correlation between molarity and mass-loss, as seen in Figure 6b, with a correlation coefficient R^2^ = 0.76; this indicates that the variation in removal is explained by the molarity in 76.3% of cases.

Another important correlation for the HCl surface treatment is that between loss of mass and water absorption, with a value of r = −0.683; this indicates that there is a moderate correlation between loss of mass and water absorption, the relationship is negative because, if the loss of mass increases, the water absorption decreases, with no linear correlation between the two variables. (*p*-value = 0.05). The analysis concludes that factors such as particle size/soaking time (r = −0.007, *p*-value = 0.0975) are not significant correlations in the removal of mortar from the RCA.

On the other hand, Pearson’s correlations for H_2_SO_4_ treatment between molarity and water absorption (r = 0.055, not linearly associated between them—*p*-value = 0.00), time and molarity (r = 0.022, *p*-value = 0.022 linearly related), and time and mass-loss (r = −0.170, *p*-value = 0.015 linearly related) have a very low degrees of correlation (Table 4). The two factors that have a reasonable level of correlation are the loss of mass and molarity (r = 0.563), and the negative correlation between loss of mass and water absorption (r = −0.992). Neither of these is linearly related. This means that just like in HCl treatment, the variable that is most relevant for mass-loss is molarity, with a degree of variation of 56%; moreover, in this case, the water absorption is 99% correlated with the loss of mass. Figure 7 shows a weak linear correlation between molar concentration and mass-loss where the coefficient is R^2^ = 0.31.

In order to check whether the acid treatments for the removal of mortar in RCA have a considerable impact on the quality of the aggregates, the proposal of Silva et al. [44], which classifies the aggregates according to the minimum value of oven-dried density and maximum water absorption (Table 5), was considered. An analysis was made comparing the water absorption before and after the treatments of RCA with HCl using data collected in the literature. This type of acid was chosen because it had a significant amount of data from the authors, while other types of acid were evaluated because there are not enough data for analysis. In the authors’ water absorption values before RCA was treated with HCl, it is observed that the untreated RCAs are classified in the following way: 41.3% as BII, 34.4% as BI, 3.44% as AIII, 20.0% as AII, and 0.0% as AI. After the aggregates are treated, the percentages change to 13.3% as BII, 50.0% as BI, 6.66% as AIII, 10.0% as AII and 20.0% as AI, evidencing that there is a significant change in the properties of RCA when treated with HCl.

## 6. Conclusions

This paper presents a state-of-the-art review of the chemical treatment techniques for the removal of adhered mortar in order to improve the quality of RCA. Based on the review and analysis of the literature, the main conclusions of the article are:

RCA has seen improvements in its physical and mechanical characteristics when treated with solutions at low concentrations, e.g., improving water absorption, density, and RCA surface. From the literature data, it can be concluded that treatments with acids can change the quality classification of RCA for the better.

The immersion time in the acidic solution does not linearly relate to the adhered mortar removal volume. This is due to the fact that, during the removal of mortar, the sulfate ions (SO_4_^2−^) are consumed in the reaction in the first steps. The total removal of the paste present in the RCA occurs when the acidic concentrations are increased to values greater than 3 M [6].

Concrete mixes that contain treated RCA showed better mechanical behavior than those incorporating RCA without treatment. Thanks to the removal of the paste and improvement of the ITZ between RCA and the new cement paste, higher tensile strength, elasticity module, and tensile strength are achieved. In terms of durability, treated RCA mixes had better values than untreated RCA ones.

There is no strong linear relationship between the variables that control mortar removal and the final physical properties of RCA. This can be explained by the high variability of RCAs both in their physical and mechanical properties.

From the literature review carried out, it can be concluded that the use of acids as a treatment for improving the physical and mechanical properties of RCAs can be an optimal solution, depending on the environment to which the concrete will be exposed. An option for this type of treatment would be acid drainage generated by the mining industry, where there is potential for using this type of waste.

## Figures and Tables

**Figure 1 materials-15-02740-f001:**
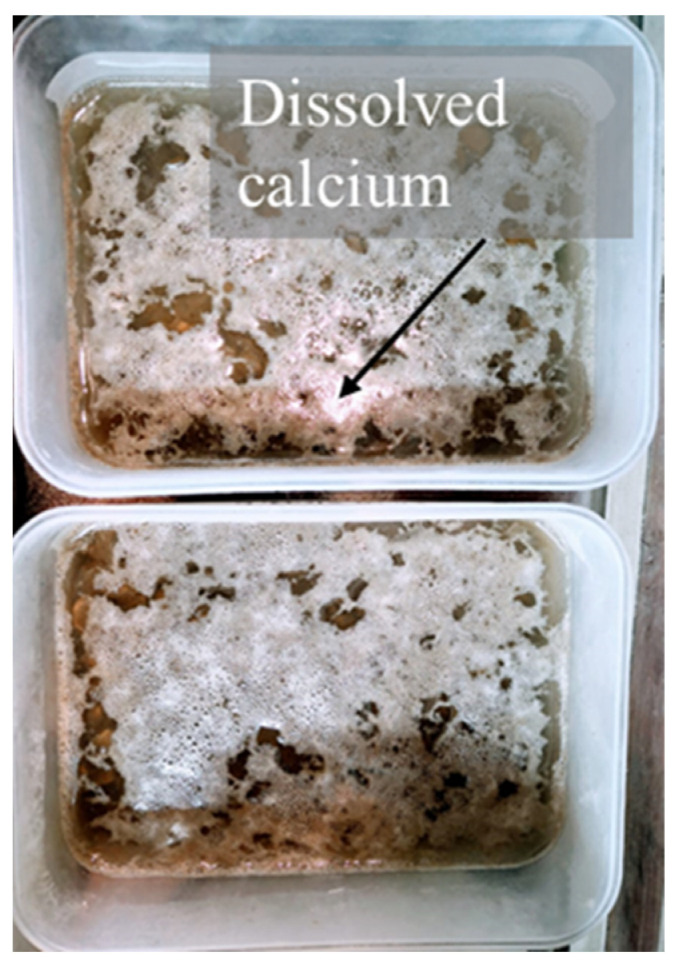
Calcium dissolved from RCA in acidic solution.

**Figure 2 materials-15-02740-f002:**
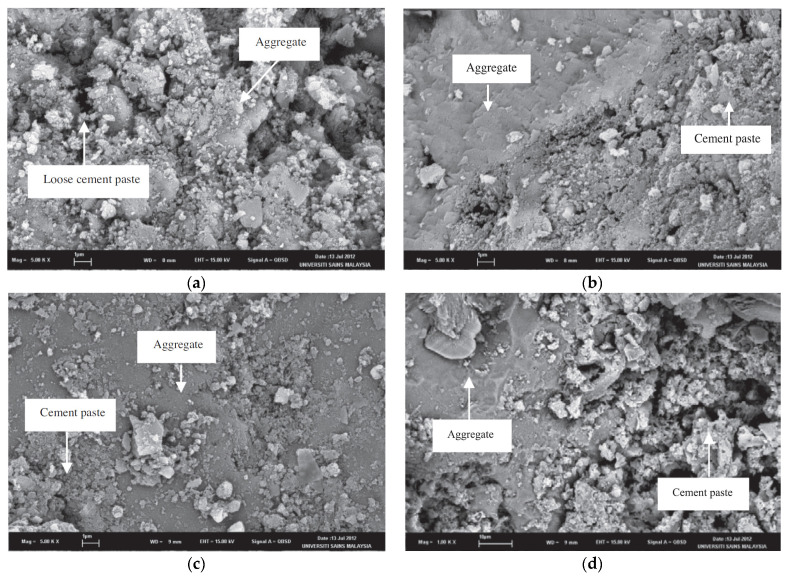
(**a**) Normal (untreated RCA); (**b**) treated RCA at 0.1 M; (**c**) treated RCA at 0.5 M; and (**d**) treated RCA at 0.8 M (Ismail and Ramli [9]).

**Figure 3 materials-15-02740-f003:**
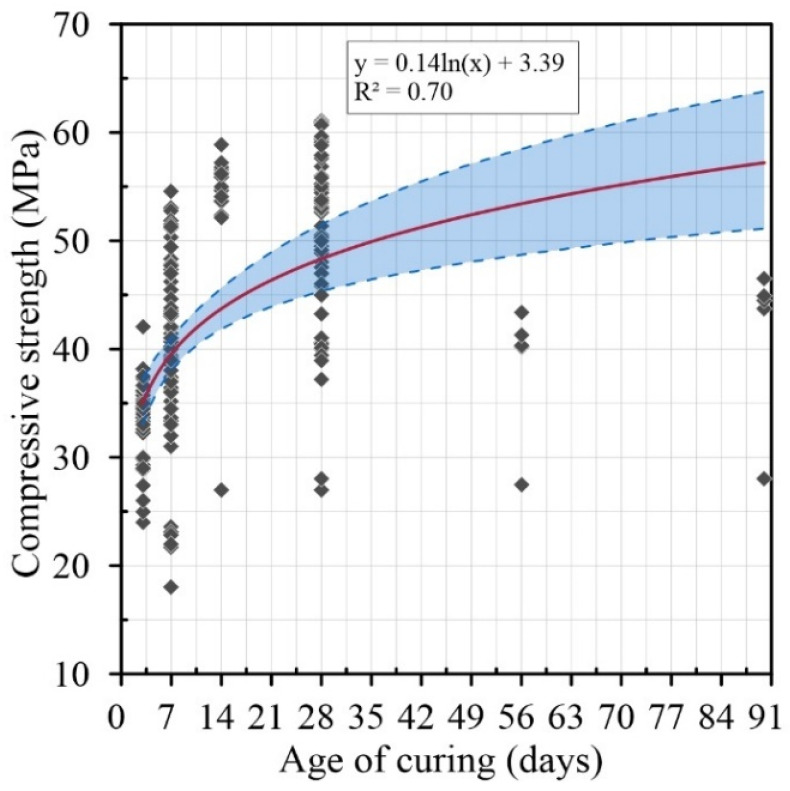
Age of curing (days) versus compressive strength with HCl treatment.

**Figure 4 materials-15-02740-f004:**
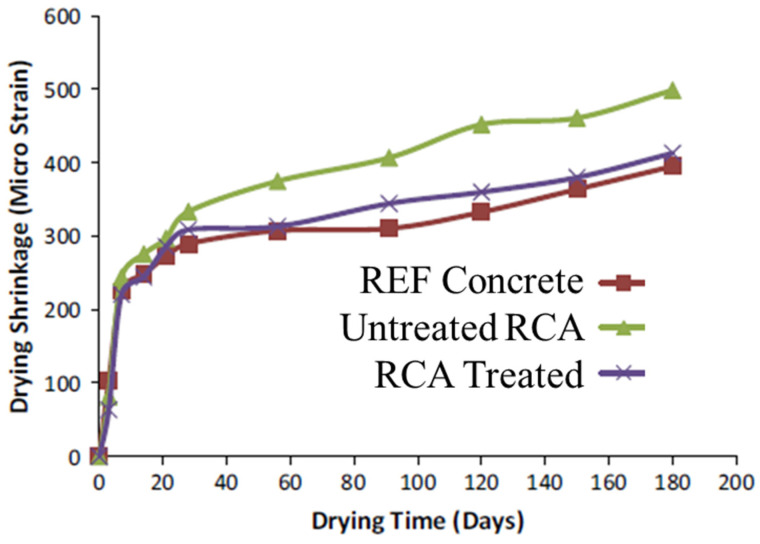
Drying shrinkage of concrete mixes versus drying time (Ismail and Ramli [10]).

**Figure 5 materials-15-02740-f005:**
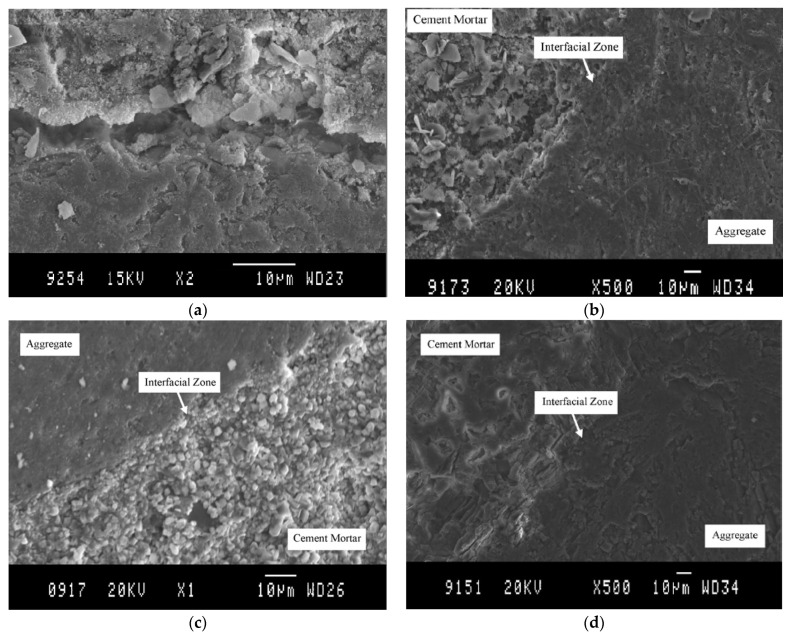
(**a**) ITZ for concrete with RCA without pre-soaking treatments; (**b**) interfacial zone for RCA treatment with HCl; (**c**) interfacial zone for RCA treatment with H_2_SO_4_; and (**d**) interfacial zone for RCA treatment with H_3_PO_4_ (Tam et al. [15]).

**Figure 6 materials-15-02740-f006:**
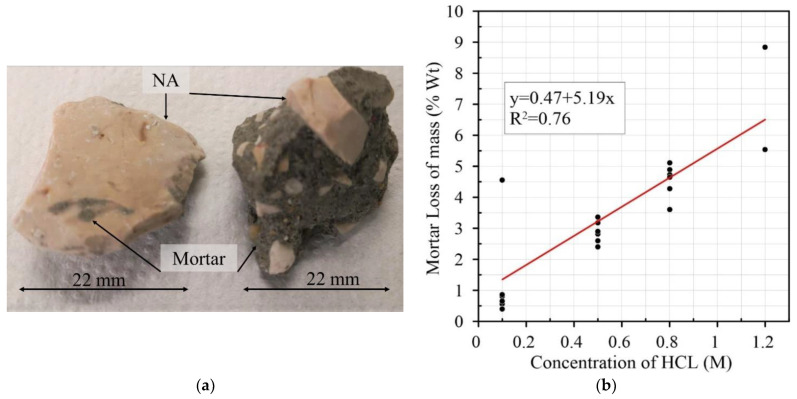
(**a**) CA particle size with different mortar contents; (**b**) relationship between concentration of HCl (M) and mortar loss of mass (% Wt) [6,8,9,10,11,13,14,15].

**Figure 7 materials-15-02740-f007:**
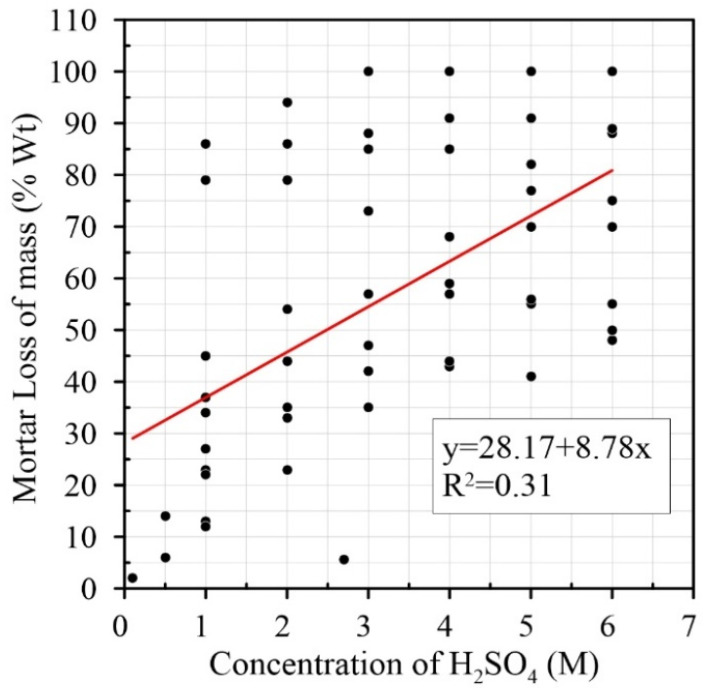
Relationship between concentration of H_2_SO_4_ and mortar loss of mass (% Wt) [6,7,13,14,15].

**Table 1 materials-15-02740-t001:** Summary of physical and mechanical properties measured in concrete and mortar mixes, and techniques reported for RCA.

Measured Parameters	Technique	References
Water absorption	Immersion of RCA in acidic HCl, H_2_SO_4_, and H_3_PO_4_	Tam et al. [15]
Water absorption, mortar content, and bulk density	Immersion of RCA in acidic H_2_SO_4_	Akbarnezhad et al. [6,7]
Water absorption, mortar content, bulk density, and microscopic analysis of the RCA	Immersion of RCA in acidic HCl	Ismail and Ramli [9,10]
Water absorption	Immersion of RCA in acidic HCl and H_2_SO_4_	Purushothaman et al. [13]
Water absorption	Immersion of RCA in acidic HNO_3_	Pandurangan [6,12]
Water absorption, mortar content, and bulk density	Immersion of RCA in acidic HCl, H_2_SO_4_ and HNO_3_	Saravankumar et al. [14]
Water absorption, mortar content, and microscopic analysis of the RCA	Immersion of RCA in acidic HCl and C_2_H_4_O_2_	Al-Bayati et al. [8]
Water absorption and mortar content	Immersion of RCA in acidic HCl and Na_2_SO_4_	Kim et al. [11]
Mortar content	Immersion of RCA in acidic HCl	Juan and Gutierrez [50]
Mortar content	Immersion of RCA in sodium sulfate (Na_2_SO_4_), magnesium sulfate (MgSO_4_), and magnesium chloride (MgCl_2_)	Abbas et al. [5,51]
Water absorption, bulk density, and specific gravity	Immersion of RCA in acidic H_2_SO_4_	Tang et al. [49]
Water absorption and apparent density	Immersion of RCA in acidic CH_3_COOH	Want et al. [20]

**Table 2 materials-15-02740-t002:** Summary of measurements of fresh-state properties, mechanical properties and durability.

Parameter	Concrete Mix	References
Density	Concrete with RCA treated with HCl and Na_2_SO_4_	Al-Bayati et al. [8]
Workability of concrete, density compressive strength, tensile strength, UPV, modulus of elasticity (E), and shrinkage	Concrete with RCA treated with HCl	Ismail and Ramli [9,10]
Workability of concrete and compressive strength	Concrete with RCA treated with HNO_3_	Pandurangan et al. [12]
Compressive strength and modulus of elasticity (E)	Concrete with RCA treated with HCl, H_2_SO_4_, and H_3_PO_4_	Purushothaman et al. [13]
Compressive strength	Concrete with RCA treated with HCl, H_2_SO_4_, and HNO_3_	Saravanakumar et al. [14]
Compressive strength, tensile strength, and modulus of elasticity (E)	Concrete with RCA treated with HCl, H_2_SO_4_, and H_3_PO_4_	Tam et al. [15]
Compressive strength, tensile strength, and modulus of elasticity (E)	Concrete with RCA treated with HCl and H_2_SO_4_	Wang et al. [20]
Compressive strength	Concrete with RCA treated with HNO_3_	Pandurangan et al. [12]
Compressive strength, chloride ion penetrability, and carbonation resistance	Concrete with RCA treated with HCl and Na_2_SO_4_	Kim et al. [11]
Compressive strength, flexural strength, and modulus of elasticity (E)	Mortar with RCA treated with HCl and H_2_SO_4_	Kim et al. [60]
Workability of concrete	Concrete with RCA treated with HCl and HNO_3_	Butler et al. [57]

**Table 3 materials-15-02740-t003:** Pearson correlation for RCA treated HCl.

	Molarity (M)	Time (days)	Size of Aggregate (mm)	Water Absorption (%)
Molarity (M)	1	-	-	-
*p*-Value	1	-	-	-
Time	−0.094	1	-	-
*p*-Value	0.633	1	-	-
Size of aggregate (mm)	0.077	−0.007	1	-
*p*-Value	0.708	0.975	1	-
Water absorption (%)	−0.140	0.235	−0.211	1
*p*-Value	0.478	0.230	0.300	1
Mortar loss (%)	0.400	−0.170	−0.368	−0.632
*p*-Value	0.050	0.415	0.077	0.000

**Table 4 materials-15-02740-t004:** Pearson correlation for RCA treated with H_2_SO_4_.

	Molarity (M)	Time (days)	Water Absorption (%)
Molarity (M)	1	-	-
*p*-Value	1	-	-
Time	−0.300	1	-
*p*-Value	0.022	1	-
Water Absorption (%)	0.055	0.135	1
*p*-Value	0.889	0.730	1
Mortar loss (%)	0.563	−0.170	−0.992
*p*-Value	0.000	0.015	0.000

**Table 5 materials-15-02740-t005:** Physical property requirements for the proposed classes [44].

Aggregate Class	A			B			C			D
	I	II	III	I	II	III	I	II	III	
Maximum water absorption (%)	1.5	2.5	3.5	5	6.5	8.5	10.5	13	15	No limit

## Data Availability

No new data were created or analyzed in this study. Data sharing is not applicable to this article.

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
