# Peer review of "Improvement of the Quality of Recycled Concrete Aggregate Subjected to Chemical Treatments: A Review"

_materials, 2022, doi:10.3390/ma15082740_

Round 1
Reviewer 1 Report
Dear Editor,
Thank you very much for the opportunity to review the manuscript "Improving the Quality of Chemically Treated Recycled Concrete Aggregate: A Review"
In this work the authors present the current state of the scientific art on treatments for the removal of adherent paste with acid solutions on the Aggregate of Recycled Concrete (RCA) and their influence on the mechanical properties and durability of the concrete produced with aggregate concrete. recycled (RAC). In addition, they present the results of statistical analysis, using Pearson's correlation, to determine the linear relationship of the main factors involved in pulp removal in RCA.
In details:
The introductory section is supported by a good list of references and the main objectives of the work are well indicated.
The second paragraph appropriately describes the state of the art regarding the Chemistry of Acid Treatment in RCA; the third and fourth paragraphs describe in an adequate and detailed way the main properties of RCA after treatment with acid solutions, collecting the literature data, giving a complete picture of the knowledge that exists in this area.
The understanding concerning the statistical treatment of data relating to RCA is very interesting.
In the concluding paragraph the authors summarize briefly what is stated in their manuscript and end with the affirmation that the high variability of RCA both in their physical and mechanical properties means that there is no strong linear relationship between the variables that control the removal of the mortar and the final physical properties of RCA. They also conclude by stating that an option for this type of treatment would be acid drainage generated by the mining industry, where the possibility of using this type of waste exists.
In my opinion, this article is a good manuscript, presented in a rational way, the only objection I would like to raise concerns the use of acid attacks for the treatment of these materials. In addition to presenting another environmental impact, one wonders whether the authors have evaluated the possibility that the treatment with acid attacks does not act only on the cement paste but also on the aggregate clasts, especially if they have a carbonate composition.
Author Response
Thank you very much for each of your observations and comments. In response to your concern, as proposed by other authors, the incorporation of hydrochloric acid in low concentrations and a short immersion time is expected to have low impacts on calcareous aggregate. Therefore, hydrochloric acid would only affect the superficial layers where the cementing paste is located.
Reviewer 2 Report
1.Recheck "The best water absorption performance was achieved at RCA that were treated by using 1% acetic acid. Want et al. explained that this can be attributed to the fact that more hydration products, and possibly some NA were dissolved by the acetic acid at higher concentration. As result, more pores were produced in the treated RCA samples, leading to higher water absorption. "
2.Need to discuss more " A summary of the different mechanical parameters that have been determined so far for concrete made with RCA treatment, and the type of acid solution is shown in Table 2."
3.Need to rewrite "From the bibliographical analysis it can be concluded that the use of acids to improve the properties can be a solution in certain environments. An option for this type of treat ment would be acid drainage generated by the mining industry. where there is a potential for using this type of waste. "
Author Response
Reviewer #2:
Comment #1
Recheck "The best water absorption performance was achieved at RCA that were treated by using 1% acetic acid. Want et al. explained that this can be attributed to the fact that more hydration products, and possibly some NA were dissolved by the acetic acid at higher concentration. As result, more pores were produced in the treated RCA samples, leading to higher water absorption. "
Answer:
Thank you very much for the observation. This aspect was correct addressed and explained in more detail.
Comment #2
Need to discuss more " A summary of the different mechanical parameters that have been determined so far for concrete made with RCA treatment, and the type of acid solution is shown in Table 2."
Answer:
Table 2 provides the readers with a summary of the properties researched by the different authors. A more thorough analysis of those results has been conducted throughout the text.
Comment #3
Need to rewrite "From the bibliographical analysis it can be concluded that the use of acids to improve the properties can be a solution in certain environments. An option for this type of treatment would be acid drainage generated by the mining industry. where there is a potential for using this type of waste."
Answer:
As suggested, the authors improved the wording of the paragraph.
Reviewer 3 Report
A complete re-writing of the manuscript would be necessary, starting from the chemistry of RCA acid treatments whose description is confused, repetitive and, at times, incorrect. A better organization of the section dealing with the properties of treated recycled aggregates would be desirable so that the eventual differences among the properties of RCA subjected to the various kinds of acid treatments will be put in evidence. The effects of the different RCA treatments in comparison to non-treated RCA should also be better evidenced in the section dealing with the properties of concrete containing recycled aggregates. Obviously, concretes with the same composition are necessary for the comparison.
Revision for the English language is also required.
Some specific remarks are reported below.
Reactions from (1) to (8) represent the possible mechanisms for hydrated cement degradation. The authors should specify if these mechanisms were considered as responsible for the mortar removal from recycled aggregates by means of acid treatments.
Formal aspects involving the nomenclature of chemical species should be considered. For example, the use of “Acid X” in Equation (1) and Equation (9) is not acceptable as a chemical formula. This is also considering that cement chemistry notation has been used for the other species. The calcium salt (CX) is a reaction product (Eq. (1)), not a by-product (line 59).
Sulphate ions are repeatedly indicated as “(H+)” (lines 73, 81, 151). C4A is reported as a reagent in equations (5) and (6) instead of C3A.
The chemical reaction reported in Equation (9) is not clear: what is CaA2? Furthermore, assuming the acid being monoprotic (HX) the reaction would give the calcium salt (CaX2), CO2 and water.
The statement “The acidolysis process is predominant in the attack of sulphates” (line 90) is not clear.
The content of Table 1 is not clear. I suppose the techniques reported in the second column represent those for RCA treatments instead of those for determining the properties in the first column.
Author Response
Comment #1
A complete re-writing of the manuscript would be necessary, starting from the chemistry of RCA acid treatments whose description is confused, repetitive and, at times, incorrect. A better organization of the section dealing with the properties of treated recycled aggregates would be desirable so that the eventual differences among the properties of RCA subjected to the various kinds of acid treatments will be put in evidence. The effects of the different RCA treatments in comparison to non-treated RCA should also be better evidenced in the section dealing with the properties of concrete containing recycled aggregates. Obviously, concretes with the same composition are necessary for the comparison.
Answer:
As suggested, the authors performed a complete revision of the document. Several parts of the document were rewritten.
Comment #2
Revision for the English language is also required.
Answer:
As suggested, the authors performed a complete revision of the document. Several parts of the document were rewritten.
Comment #3
Some specific remarks are reported below.
Answer:
NA
Comment #4
Reactions from (1) to (8) represent the possible mechanisms for hydrated cement degradation. The authors should specify if these mechanisms were considered as responsible for the mortar removal from recycled aggregates by means of acid treatments.
Answer:
This aspect been clarified.
Comment #5
Formal aspects involving the nomenclature of chemical species should be considered. For example, the use of “Acid X” in Equation (1) and Equation (9) is not acceptable as a chemical formula. This is also considering that cement chemistry notation has been used for the other species. The calcium salt (CX) is a reaction product (Eq. (1)), not a by-product (line 59).
Answer:
These errors were corrected.
Comment #6
Sulphate ions are repeatedly indicated as “(H+)” (lines 73, 81, 151). C4A is reported as a reagent in equations (5) and (6) instead of C3A.
Answer:
These errors were corrected.
Comment #7
The chemical reaction reported in Equation (9) is not clear: what is CaA2? Furthermore, assuming the acid being monoprotic (HX) the reaction would give the calcium salt (CaX2), CO2 and water.
Answer:
Thank you very much for your valuable contribution. As you mention, the use of CaA2 is inappropriate. These aspects were clarified.
Comment #8
The statement “The acidolysis process is predominant in the attack of sulphates” (line 90) is not clear.
Answer:
The authors find this comment very appropriate. The main idea in line 90 was adjusted.
Comment #9
The content of Table 1 is not clear. I suppose the techniques reported in the second column represent those for RCA treatments instead of those for determining the properties in the first column.
Answer:
The title of table 1 was modified as the previous title caused confusion and did not correctly represent the content.
Round 2
Reviewer 2 Report
Accept